# The magnitude of anemia and associated factors among adult diabetic patients in Tertiary Teaching Hospital, Northern Ethiopia, 2019, cross-sectional study

**Nigus Alemu Hailu** [1]*, **Tesfaye Tolessa**[2], **Zenawi Hagos Gufue**[3], **Etsay Weldekidan Tsegay**[4], **Kidanemaryam Berhe Tekola**[5]

1 Biomedical Sciences Department, College of Medicine and Health Sciences, Adigrat University, Adigrat, Ethiopia, 2 Medical Physiology Department, College of Health Sciences, Addis Ababa University, Addis Ababa, Ethiopia, 3 Department of Public Health, College of Medicine and Health Sciences, Adigrat University, Adigrat, Ethiopia, 4 Pharmacy Department, College of Medicine and Health Sciences, Adigrat University, Adigrat, Ethiopia, 5 Nutrition Department, College of Health Sciences, Mekelle University, Mekelle, Ethiopia

* Nigusalemu2@gmail.com

**Data Availability Statement:** All relevant data are within the manuscript and its Supporting Information files.

## Abstract

### Background

Patients with Diabetic Mellitus are at higher risk of different complications. Many previous studies show that anemia among diabetic patients is poorly diagnosed.

### Objective

This study aimed to assess the magnitude and associated factors of anemia among adult diabetes patients having regular follow up at the diabetic clinic of Ayder Comprehensive Specialized Hospital, Tigray, 2018/19.

### Methods

This study was conducted the Diabetic clinic of Ayder comprehensive specialized hospital, Tigray regional state, Northern Ethiopia from January to March 2019. A systematic random sampling technique was used to select study participants. About 5 ml of venous blood was collected by experienced laboratory technologists under a complete aseptic technique. Two ml of the venous blood was used for hemoglobin determination. And three ml of the venous blood was used without any anticoagulant for creatinine determination. The association of variables was assessed using bivariate and multivariable analysis in the logistic regression model with p-value, odds ratio, and 95% CI in the SPSS version 24 software.

### Results

From a total of 262 diabetes patients, forty-seven (17.9%) were found to be anemic (6.7% males and 11.5% females). Among the related factors, residency (Adjusted Odds Ratio, 7.69, 95% CI, 2.060, 28.69, p = 0.002,), age of the patients (Adjusted Odds Ratio, 4.007,

**Funding:** The corresponding author gets fund from Addis Ababa University (internal organization) but the funder had no role in study design, data collection, analysis and decision to publish or preparation of the manuscript.

**Competing interests:** The authors have declared that no competing interests exist.

95%CI, 1.53–10.51, p = 0.005,) and sex (Adjusted Odds Ratio, 3.434, 95% CI, 1.582, 7.458, p = 0.042,) were significantly associated with anemia.

## Conclusion

According to this study, the magnitude of anemia is high among diabetic patients. Occupation of the participants, residency, HIV status, being female, and age was significantly associated with anemia.

## Background

Diabetes mellitus (DM) is a heterogeneous metabolic disorder characterized by the presence of hyperglycemia due to impairment of insulin secretion, defective insulin action, or both, which requires continuous medical care with multifactorial risk-reduction strategies beyond glycemic control. The chronic hyperglycemia of patients with diabetes mellitus is associated with relatively specific long-term microvascular complications affecting the eyes, kidneys, and nerves, as well as an increased risk of cerebrovascular disease [1,2].

Anemia is one of the preventable complications of DM, but it is less focused and estimated by the diabetes care providers [3]. Globally, Iron deficiency is the most common cause of anemia but other nutritional deficiencies (including folate, vitamin B12, and vitamin A), acute inflammation, chronic inflammations, and inherited or acquired disorders can also cause anemia by reducing hemoglobin synthesis, red blood cell production, or red blood cell survival '[2].

Hemoglobin concentration alone cannot be sufficient to diagnose iron-deficiency anemia, but it can provide general information on the magnitude and severity of anemia [4]. The cutoff points to diagnose anemia is <12 g/dl and <13 g/dl of Hemoglobin for females and males respectively [2]. DM and anemia are closely associated and diabetic patients tend to develop anemia because of different factors, including free radicals, pro-inflammatory cytokines, and renal disease [5].

The prevalence and incidence of diabetes has been reported to vary among populations and among age groups of the same population. According to the International Diabetic Federation 2015 estimate, and WHO 2016 report, the worldwide prevalence of DM is 1 in 11 adults, and 422 million people were living with DM. There were also 5 million deaths related to it [6]. Diabetes among adults is increased from 4.7% in 1980 to 8.5% in 2014 [7]. From this, Africa accounts for 14.2 million diabetic cases and this is expected to rise to 34.2 million by the end of 2035 [6].

DM affects virtually every system of our body. Most of the complications develop from hyperglycemia which in turn, increases the generation of oxygen-derived free radicals. The free radicals will cause further vascular complications and damage [8]. Anemia is one of the common complications of diabetes mellitus. It is poorly diagnosed and neglected disorder among diabetic patients. This is due to poor screening of their hematologic conditions. This will expose diabetic patients to different unwanted complications and it will be difficult to treat at its advanced stage [9].

Globally, anemia affects more than 1.62 billion people (24.8% of the global population) and it is responsible for 8.8% of the total disabilities globally. South Asia and Sub-Saharan Africa are among the highly burdened zones, while the westerns are least affected. Ethiopia is one of the highly affected Sub-Saharan countries by anemia [10]. Anemia is the second target of

global nutrition for 2025 [9]. Anemia has a significant negative impact on the overall global and national development [11].

The precise magnitude of anemia among chronic disease patients is difficult to ascertain because many of the patients with DM and other chronic diseases are not screened and investigated sufficiently to establish the diagnosis of anemia [10]. Anemia among diabetic patients is an augmenting risk factor for further complications like cardiovascular diseases, and increased morbidity and mortality. Diabetic Patients who are anemic are prone to develop left ventricular hypertrophy which eventually leads to full-blown cardiovascular disease and chronic renal disease [12].

In Ethiopia, there are only a few researches done on the association of hematologic profiles and renal disease among diabetic patients. Additionally, in Ethiopia anemia screening among diabetes patients is not practiced during their follow-ups. Therefore, anemia can put diabetes patients at high risk of hospitalization and premature death. But there is an information gap on the exact magnitude of anemia among adult diabetic patients. The information is critical to ensure optimum delivery of comprehensive follow-up services.

Therefore, this study was aimed to assess the magnitude and associated factors of anemia among diabetes mellitus patients at Ayder Comprehensive Specialized Hospital in 2018/19. So, there is a need to conduct a research to realize feasible solutions and to inform clinicians on successful monitoring anemia status among diabetes patients and to prevent the possible complications at their early stage. Besides, this study will help clinicians, policymakers, and health planners in designing the best and appropriate early anemia screening and approaching strategies.

## Materials and methods

### Study design, period and area

Institutional based cross-sectional study was conducted from January to March 2019. The study was conducted at the diabetic clinic of Ayder Comprehensive Specialized Hospital, Tigray Regional State, Northern Ethiopia, which is found 783 Kilometers North of Addis Ababa, the capital city of Ethiopia. It is a public teaching hospital and research center in Northern Ethiopia, rendering its referral and specialized medical services to more than 9 million populations in its catchment areas. It stands as the second-largest hospital in the nation with a total capacity of about 500 inpatient beds in all departments and other specialty units, with more than 170,000 patient flows per year [13].

### Study participants

The study was conducted on 262 adult diabetic patients. The target population is all adult diabetes patients while the adult diabetic patients who visited the clinic during the study period and fulfilling the inclusion criteria were considered as the study population. All confirmed outpatient adult diabetes patients who had regular follow up at the diabetic clinic of Ayder Comprehensive Specialized Hospital during the study period were included in the study. The diabetic patients were diagnosed based on the WHO 2006 criteria of DM diagnosis. Adult diabetic patients who had confirmed cancer cases or taking immunosuppressive medications causing anemia, pregnant mothers, admitted patients, critically ill patients and those who had known hearing problem that and then cannot able to give informed consent; those who were taking anthelminthic medication in the last three months were excluded from the study. The medications were checked from the patient's chart.

## Sample size and sampling technique

The sample size was determined using single population proportion formula using STAT CALC menu of Epi info version 7.2.2.6 software (center for diseases control and prevention, Atlanta, USA) with the assumptions of two-sided significance level ($\alpha$ = 5%), 95% confidence level, considering 19% prevalence of anemia among adult diabetes patients [14], and accordingly, the calculated sample size was 236. Adjustment for non-response rate was made by taking 10% and then the final sample size was 262. A systematic random sampling technique was used to select study participants. There were a total of 2400 diabetic patients with a regular follow up during the study period. And using systematic random sampling, every 9th patient was selected.

## Data collection and laboratory methods

A pre-tested structured interviewer-administered questionnaire was used to collect data on socio-demographic characteristics of patients and their associated risk factors, which have been developed after reviewing different kinds of literature and previous similar studies by the authors. The questionnaire was prepared first in English and then translated to the local language (Tigrigna) and then back to English to maintain its consistency. Data were collected by two experienced nurses at the diabetic clinic of Ayder comprehensive specialized hospital and one supervisor with masters in diabetes care, who was recruited from the same Hospital. The data collection period was from January to March 2019.

Anthropometric measurements like height and weight were measured using the nearest centimeter without shoes and to the nearest 0.1kg respectively. Body mass index was calculated by dividing the weight in Kg by the square of height in meter. Waist circumference was measured midway between the lowest rib and iliac crest. Blood Pressure was measured using the sphygmomanometer from the upper arm after a patient sat for at least 5 minutes by qualified personnel. The hand was at heart level to minimize the gravity effect during blood pressure measurement, and all the anthropometric and BP measurements were measured twice, and the average value was taken.

A total of 5 ml venous blood was collected by an experienced laboratory technologist under a complete aseptic technique and from this, 2 ml venous blood was put into a test tube coated with EDTA anticoagulant for hemoglobin determination. The collected whole blood was properly mixed and put into an SYSMEX CBC machine for hemoglobin and hematocrit determination. The performance of the hematology analyzer was controlled by running quality control material alongside the study participant's sample. For creatinine determination, 3 ml of the venous blood was used by a test tube without any anticoagulant and the whole blood was allowed to clot for 20 minutes. The separated serum was put in to fully automated chemistry machine to determine creatinine.

**Data quality assurances.** To assure the quality of the data, a properly designed data collection tool was prepared before starting the actual data collection process and closed supervision was given by the principal investigator during data collection, collected data was reviewed by the principal investigator, any problem faced at the time of data collection was discussed and corrective measures were made immediately. In the laboratory aspect, quality assurance checks were performed daily according to the laboratory's protocol. Standard operating procedures were followed during specimen collection and processing. All reagents used were checked for their expiry date and prepared according to the manufacturer's instructions.

**Data processing and analysis.** The data was entered into Statistical Package for the Social Sciences (SPSS) version 24.0 for windows (Chicago, IL, USA). Descriptive statistics were presented as means and standard deviations, while categorical variables were presented in

frequencies and percentages. Logistic regression was computed to assess the statistical association. Bivariate analysis was done to check the existence of crude association and to select candidate variables, those variables which were clinically important and those which had a P-value of < 0.25 in the bivariate analysis were selected to enter into the final model.

A multivariable binary logistic regression model was used to identify the independent factors associated with anemia. The summary measures of estimated crude and adjusted odds ratios with 95% confidence interval were presented and P-value <0.05 was used to declare statistical significance.

**Ethical considerations.**   Ethical approval was obtained from the research and ethical committee of the Department of Medical Physiology, School of Medicine, College of Health Sciences, Addis Ababa University. The chief executive director of Ayder Comprehensive Specialized Hospital and head of the diabetic clinic of the hospital were informed about the objective of the study and written permission was obtained before starting data collection. All participants were asked to provide written informed consent if they were able to write and for those who cannot write they were asked to use inked thumbprint on the consent form in the presence of an independent witness. Each respondent was informed about the objective of the study and they were assured of the confidentiality, risks, and benefits of the study procedures.

**Dissemination of findings.**   This finding was defended at the Department of Medical Physiology, School of Medicine, College of Health Science, Addis Ababa University, and submitted to the School of graduate studies of Addis Ababa University, principal advisor of the thesis, Ayder Comprehensive Specialized Hospital medical director office, and other concerned bodies. The result is also to be disseminated through workshops, seminars, and published in an international, professional high impact journal.

## Results

### Socio-demographic characteristics of study participants

A total of 262 adult diabetes patients were included in this study; from whom, 134 (51.1%) were males while 128 (48.9%) were females. The response rate was 100%. The minimum age of the participants was 18 years, while the maximum was 84. About two hundred (76.3%) of the participants were between the age of 18 and 60 years old (Table 1).

**Clinical profile of the participants.**   Almost all the participants, 232 (85.5%) had good adherence to their medication and 168 (64.1%) patients had metformin as their first-line medication, while 35.1% of them were taking insulin as a first-line medication. The average stay with diabetes mellitus from the first time of diagnosis was about 8.7 ±6.8 years. Twenty-seven (10.3%) participants had chronic eye disease and twenty-nine (11.1%) had arthritis. Ten (3.8%) patients had confirmed retroviral infection. Fourteen (5.3%) patients had an un-specified chronic disease that they could not rule out, other than the above-mentioned diseases. In this study, about 26% of type-2 DM patients had a BMI of greater than 25 Kg/m$^2$, while only 1.9% of type-1 DM patients had a BMI greater than 25 Kg/m$^2$.

**Biochemical profiles of the participants.**   A waist circumference of the patients was measured to see the presence or absence of central obesity. Majority (88.1%) of males had a waist circumference of less than 102 cm, while16 (11.9%) males had waist circumference above 102 cm. One hundred one (78.1%) female participant had a waist circumference of less than 88 cm, while 28 (21.9%) of them had a waist circumference of above 88 cm.

Majority (80.5%) of the participants had systolic blood pressure between 90 and 140 mmHg. Fifty (19.1%) patients had systolic blood pressure greater than 140 mmHg. Similarly (83.2%) of the participants had diastolic blood pressure between 60 to 85 mmHg and 42 (16%) had diastolic blood pressure greater than 90 mmHg. One hundred twenty-six (52.9%) of the

**Table 1. Socio-demographic characteristics of diabetes mellitus patients in Ayder Comprehensive Specialized Hospital, 2019.**

| Variable | Category | Frequency | Percent (%) |
|---|---|---|---|
| Educational status | Not educated | 120 | 45.8 |
| | Primary school | 37 | 14.1 |
| | Secondary school | 36 | 13.7 |
| | Above secondary school | 69 | 26.3 |
| | Total | 262 | 100 |
| Occupation | Governmental | 70 | 26.7 |
| | Farmer | 67 | 25.6 |
| | Self-employed | 125 | 47.7 |
| | Total | 262 | 100 |
| Ethnicity | Tigray | 260 | 99.2 |
| | Afar | 2 | 0.8 |
| | Total | 262 | 100 |
| Income level | High | 35 | 13.4 |
| | Medium | 170 | 64.9 |
| | Low | 57 | 21.8 |
| | Total | 262 | 100 |

participants had a fasting blood glucose of greater than 160 mg/dl, while 109 (45.8%) had to fast blood glucose level between 70 and 160 mg/dl. A majority (65.3%) of the patients had a normal range of BMI (18.5 and 24.9 kg/m$^2$), while seventy patients (27.7%) had a BMI of above 25 kg/m$^2$. Besides, 11 patients had a BMI of <18.5 kg/m$^2$.

The prevalence of anemia among patients with a normal range of BMI was about 17%. Underweight patients have a 35.3% prevalence of anemia. Of the total anemic patients, 12.8% were underweight. From the total participants, One hundred sixty-nine (77.1%) patients had creatinine levels between 0.5 and 1.2 mg/dL, while 93 (35.6%) had creatinine levels of above 1.2 mg/dL.

From the total 47 anemic cases in this study, nearly half of the patients (46.8%) had a Cr level of greater than 1.2 mg/dL. The distribution of anemia differs from the sex of the participant, where females have a higher prevalence of anemia. In addition, the prevalence of anemia is higher among DM patients with the renal disease compared to DM patients with normal renal function.

Bivariable analysis of anemia showed that age, sex, residency area, occupational status, type of diabetes mellitus, hypertension status, and HIV infection were found to be candidates for the next model, the multivariable logistic regression model. The significance level for bivariate analysis was considered less than 0.25. The variables which were a candidate in the bivariable analysis entered into multivariable logistic analysis for identifying the most independent predictors of anemia.

The multivariable logistic regression analysis derived from binary logistic regression analysis showed that the area of residency, occupational status, HIV status, sex, and age of participants was found to be statistically significant to associate with anemia among adult diabetes patients. Even though eGFR is not statistically significant, clinically it is an important predictor of anemia among DM patients. The multivariable analysis of the candidate variables is summarized as follows, (Table 2)

## Discussion

Several studies have shown an increased risk of anemia among diabetic patients [15]. In this study, the number of male and female participants was similar (51.1%) and (48.9%)

**Table 2. Multivariable analysis of anemia in diabetes mellitus patients.**

| Variable | | Anemia | | P-VALUE | AOR | 95%CI AOR | |
|---|---|---|---|---|---|---|---|
| | | No | Yes | | | lower | Upper |
| | Residency area | | | **0.002***  | | | |
| | Urban | 175 | 29 | | 1 | | |
| | Rural | 40 | 18 | 0.002 | 7.7 | 2.06 | 28.67 |
| | Occupational status | | | **0.047***  | | | |
| | Governmental | 55 | 15 | | 1 | | |
| | Farmer | 52 | 15 | 0.025 | .19 | 0.046 | .159 |
| | Private | 108 | 17 | 0.99 | .48 | .205 | 1.12 |
| | HIV status | | | **0.023***  | | | |
| | Non-reactive | 62 | 15 | | 1 | | |
| | Reactive | 6 | 4 | 0.039 | 4.9 | 1.08 | 22.3 |
| | Unknown | 147 | 28 | 0.26 | .64 | .297 | 1.39 |
| | Hypertension status | | | | | | |
| | Yes | 84 | 14 | 0.199 | 0.54 | 0.215 | 1.377 |
| | No | 131 | 33 | | 1 | | |
| | Type of DM | | | 0.095 | | | |
| | Type 1 | 44 | 14 | | 1 | | |
| | Type 2 | 171 | 33 | 0.095 | 0.496 | 0.218 | 1.128 |
| | Sex of participant | | | **0.002***  | | | |
| | Male | 117 | 17 | | 1 | | |
| | Female | 98 | 30 | 0.042 | 3.43 | 1.58 | 7.46 |
| | Age of participant | | | **0.014***  | | | |
| | 18–60 years | 167 | 33 | | 1 | | |
| | >60 years | 48 | 14 | 0.005 | 4.007 | 1.53 | 10.51 |

respectively. The minimum age of the participant was 18 years, while the maximum was 84 years. This happened because the study is conducted exclusively in adult diabetic patients, and the minimum cut off point of age to consider as an adult is 18 years in this setting. Since the study was conducted in adult diabetes patients, type-2 DM patients were high in number than type-1DM patients. This is because type-2 diabetes mellitus is more common in the adult population than type-1 diabetes [7].

Out of the total participants in this study, 18% of the diabetes patients had anemia. This finding is similar to a previous study done in England, which showed a 16.1% prevalence of anemia among diabetes patients [16]. This is also consistent with a study conducted in China which showed a 22.8% prevalence of anemia among diabetes patients [17]. In diabetic mellitus patients, concurrent infections like HIV, hookworm, and TB are common causes of anemia [18]. In this study, about 3.8% of the patients had confirmed retroviral infection and those reactive patients had 1.7 times increased chance of to be anemic compared with non-reactive diabetic patients.

A study conducted in Australia [19] showed a 19% prevalence of anemia among diabetes patients. This is similar to our finding (18%). This similarity may be explained by the inclusion of similar characteristics of patients. But our result is slightly higher than another study in Australia, which showed a 14.3% prevalence of anemia among diabetic patients [20]. This difference is explained by the composition of our study subjects. In this study, the study subjects were both type-1 and type-2 diabetic mellitus patients while the mentioned study was conducted only in type-1 diabetic patients. The other possible reason for the higher result in this

study is the inclusion of a higher number of diabetes patients with renal failure, while that study includes only small number of patients with renal failure.

The result of this study is slightly higher than from a study conducted in Israel, which reported that the prevalence of anemia among diabetic patients was 10.8%. Compared to this report, our result is higher. This may be due to the inclusion of diabetes patients with different degrees of renal impairment and small sample size, unlike that study which includes only diabetic patients with normal renal function and a large sample size [15]. Those diabetic patients with hypertension had 3 times increased risk of anemia as compared to the normotensive diabetic group. This is in line with a study conducted in Korea. This may be due to the effect of vascular complications and anti-hypertensive drugs [21].

Our result is also in agreement with a study conducted in Nigeria, with a general prevalence of 15.3% in diabetes patients. Our result also shows that the prevalence of anemia was high (29.2%) with an AOR of 4.007, among patients of advanced age (>60 years), compared to patients below 60 years with (19.8%) prevalence. This is consistent with a study conducted in Nigeria [22]. This was expected because age is associated with anemia irrespective of the disease status.

The current study's finding revealed that rural residents of diabetes patients have a higher chance (AOR, 7.687 p = 0.002 CI, 2.060, 28.689) of getting anemia as compared to urban residents. This can be explained by the better level of living standards and increased health care seeking behavior in the urban. Reactive HIV status also shows a significant association with anemia among diabetes patients, with (p = 0.039 AOR, 4.92, 95%CI, 1.09–22.29) compared to non-reactive diabetes patients. HIV infection can cause anemia as it is a chronic catabolic disease and some of the highly active anti-retroviral drugs may even worsen the state of anemia.

In contrast to our study, research conducted in Gondar, Ethiopia [23] reveals that hematologic indices, like Hct, Hb, and WBC are increasing in diabetes patients. The difference observed in our result might be due to the indirect effect of insulin resistance on RBC hematopoietic precursor cells [24]. However, our study is in agreement with another study conducted in Finote Selam Hospital [25], which reported that 19.2% of diabetes patients had anemia. This concordance can be explained by a similar geographical location and living standard among diabetic patients.

Our study showed that 34.7% of the participants have impaired renal function, with 22.5%, 12.2% of them having eGFR of between 60 and 90, and less than 60 ml/min/1.73 $m^2$ respectively. This result is in agreement with a study conducted in China which showed 36% of the patients had impaired renal function using eGFR [17]. Similarly, this result is in line with a study conducted in Gojam, Finote Selam Hospital, Ethiopia, which showed 28.9% renal impairment among diabetic patients [25].

Interleukin-1β (IL-1β) and tumor necrosis factor α (TNFα) inhibits the release of erythropoietin by the kidney. Then Erythropoietin-stimulated hematopoietic proliferation, in turn, will be reduced. The proliferation of erythroid progenitors is also directly inhibited by TNFα, interferon-γ (IFNγ), and IL-1β). They also augment the erythrophagocytosis process by reticuloendothelial macrophage [15]. Bone marrow response is directly related to the activation of macrophages and the release of inflammatory cytokines, particularly IL-1, IL-6, tumor necrosis factor (TNF α), and interferon-gamma (INF γ) which act by inhibiting the proliferation of erythroid precursors and therefore inhibit erythropoiesis. Furthermore, the suppressive action of these cytokines on erythropoiesis-stimulating overcomes the action of EPO resulting in decreased bone marrow response to EPO and erythropoiesis [26].

Anemia in diabetes patients plays a pervasive role in the development of microvascular complications, including cardiovascular complications and vice versa [17]. Besides, anemia

hurts the sense of wellness, decrease work productivity, affects the quality of life, increases morbidity, mortality, and hospital stay.

The strength of this study is this is the first research to assess the burden of anemia among diabetic patients in the stated catchment area, and it tried also to assess the associated factors of anemia among diabetic patients. The limitations, it would have been more conclusive if it used a comparison group and assessed the level of their blood glucose level using hemoglobin A1c, to see the past three month's blood glucose status of the patients. But there was no fund to buy all the necessary materials and Hb A1C machine.

## Conclusion

In this study, the prevalence of anemia among adult diabetic mellitus patients is high. Therefore, anemia is a public health problem according to the finding of this research. However, it is not prioritized as one of the top problems among these patients in this study area before. The prevalence of anemia is higher among female diabetic mellitus patients in this study. In addition, old aged diabetic mellitus patients are at high risk of developing anemia compared to the young adult diabetic mellitus patients. Therefore, it is important to incorporate an anemia screening strategy in all Diabetic Mellitus patients during their regular follow-ups.

## Supporting information

**S1 Appendix. English and Tigrigna version of the questionnaire and the data set in zipped form.**
(ZIP)

## Acknowledgments

We are grateful to Mekelle University for their technical support to this study. We would also want to thank all the participants and data collectors. Finally, we would like to thank the staff of Mekelle University, diabetic clinic, and laboratory for their cooperation during this work.

## Author Contributions

**Conceptualization:** Nigus Alemu Hailu, Zenawi Hagos Gufue, Etsay Weldekidan Tsegay, Kidanemaryam Berhe Tekola.

**Data curation:** Kidanemaryam Berhe Tekola.

**Formal analysis:** Nigus Alemu Hailu, Tesfaye Tolessa, Zenawi Hagos Gufue, Etsay Weldekidan Tsegay, Kidanemaryam Berhe Tekola.

**Methodology:** Nigus Alemu Hailu, Tesfaye Tolessa, Zenawi Hagos Gufue, Etsay Weldekidan Tsegay, Kidanemaryam Berhe Tekola.

**Project administration:** Nigus Alemu Hailu, Tesfaye Tolessa, Zenawi Hagos Gufue, Etsay Weldekidan Tsegay, Kidanemaryam Berhe Tekola.

**Writing – original draft:** Nigus Alemu Hailu, Tesfaye Tolessa, Zenawi Hagos Gufue, Etsay Weldekidan Tsegay, Kidanemaryam Berhe Tekola.

**Writing – review & editing:** Nigus Alemu Hailu, Tesfaye Tolessa, Zenawi Hagos Gufue, Etsay Weldekidan Tsegay, Kidanemaryam Berhe Tekola.

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
