## [Decision Letter · Decision Letter 0]

22 Apr 2020

PONE-D-20-02501

Magnitude of Anemia and Associated Factors among Adult Diabetes Patients in Tertiary Teaching Hospital, Northern Ethiopia, Cross Sectional Study

PLOS ONE

Dear Mr. Hailu,

Thank you for submitting your manuscript to PLOS ONE. After careful consideration, we feel that it has merit but does not fully meet PLOS ONE’s publication criteria as it currently stands. Therefore, we invite you to submit a revised version of the manuscript that addresses the points raised during the review process.

We would appreciate receiving your revised manuscript by Jun 06 2020 11:59PM. To enhance the reproducibility of your results, we recommend that if applicable you deposit your laboratory protocols in protocols.io, where a protocol can be assigned its own identifier (DOI) such that it can be cited independently in the future. For instructions see: http://journals.plos.org/plosone/s/submission-guidelines#loc-laboratory-protocols

We look forward to receiving your revised manuscript.

Kind regards,

Paolo Magni

Academic Editor

PLOS ONE

Additional Editor Comments (if provided):

The paper needs a very extensive revision.

The STROBE checklist needs to be doublechecked by the Authors.

Limitations of the study needs to be addressed in the Discussion and, concisely, in the Abstract.

All Reviewers' comments need to be addressed in full.

3. In the Discussion section, please discuss any potential limitations of your study.

"We are grateful to Addis Ababa University and Adigrat University for funding this study. We would also want to thank all the participants and data collectors. Finally, we would like to thank the staff of Mekelle University, diabetic clinic and laboratory staff for their cooperation during the work."

5. Please ensure that you refer to Figure 2-3 in your text as, if accepted, production will need this reference to link the reader to the figure.

6. Please upload a copy of Figure 5-6, to which you refer in your text on page 8-9. If the figure is no longer to be included as part of the submission please remove all reference to it within the text.

7. We note you have included a table to which you do not refer in the text of your manuscript. Please ensure that you refer to Table 3 in your text; if accepted, production will need this reference to link the reader to the Table.

Reviewers' comments:

Reviewer's Responses to Questions

**Comments to the Author**

1. Is the manuscript technically sound, and do the data support the conclusions?

Reviewer #1: Yes

Reviewer #2: Partly

2. Has the statistical analysis been performed appropriately and rigorously? 

Reviewer #1: Yes

Reviewer #2: No

3. Have the authors made all data underlying the findings in their manuscript fully available?

Reviewer #1: Yes

Reviewer #2: Yes

4. Is the manuscript presented in an intelligible fashion and written in standard English?

Reviewer #1: Yes

Reviewer #2: No

5. Review Comments to the Author

Reviewer #1: Title: Magnitude of Anemia and Associated Factors among Adult Diabetes Patients in

Tertiary Teaching Hospital, Northern Ethiopia, Cross Sectional Study

Reviewer General Comments: The authors work may have significant contributions to understand the level of anemia cases among DM patients in Ethiopia set-up. Given the limited existence of similar researches in Ethiopia, this research outcome will have importance for devising or revising health policy in the country, specifically in Ayder Hospital. Considering the strong experimental design and the aforementioned reasons, I recommend the paper to be consider for publication after a minor revision.

On the other hand, the study could have been more strong were the authors recruited a control group (non-DM subjects).

Specific comments:

1. The authors declare that they don’t have specific funding source. However, they claim in the acknowledgement section that they have funding source from Addis Ababa University and Adigrat. This conflict of depiction need to be resolved.

2. Abstract: The authors ascertained that “anemia is higher in diabetes patients attending Ayder

Comprehensive Specialized Hospital” before the study conducted which is a wrong premise. How the author explain their claim about the existence of a high incidence of anemia among DM patients before the study conducted Vs the need to conduct the study to determine the magnitude?

In general, in the abstract part the problem statement and justification should be provided unambiguously.

3. In the discussion part, the authors should support “free radicals formation due to DM” with reference(s).

Reviewer #2: 1. General comment:

- The problem the researchers tried to address is very relevant and investigation could help to give emphasis by program managers and cliniciians.

- There are a number of editorial and English grammar issues in the document

2. Abstract

- Under background section: The authors stated that anemia is higher in diabetes patients ... and its magnitude is less emphasized. This can't be relevant reason to investigate this problem. If the magnitude is known and higher as stated by the authors, what is needed just intervention not research

- The objective is presented in the abstract and not shown in the introduction. Even here the formulated objective for magnitude but the title and result the authors presented include risk factors for anemia. Therefore, either the objective should be revised or the title and result should be revised

- Under result part of abstract the presented AOR for some of the variable is very wide. Does the author understand the meaning of wide CI and take care while concluding this study?

- Under conclusion section: It is stated that "... anemia is higher"; higher than which group?

- It is also stated "...Human Immune Deficiency Virus infection... . as associted with anemia." Bu not indicated in the result section

2. Background:

- The aim of the study was not shown

3. MATERIALS AND METHODS:

3.1 Study design, period and area

- The number of DM patients and care given for the patients must be outlined so that the reader can easily understand the sampling technique, and so on

3.2 Study participants

- How did the researcher assessed whether the medication is anti-helminthic or not?

3.3 Sample size and sampling technique

2020 10:24:18 PM

- Please show the interval and sampling procedure

3.4 Data collection and laboratory methods

- Please describe when and where the data was collected from the patients

- In page four paragraph two, it is presented that "And all the anthropometric and BP measurements were measured twice." Then which one was used in the analysis?

3.5 Data processing and Analysis

- What is the outcome and independent variables. The authors are expected to describe why they used logistic regression, model fitness check?....

4 Result:

4.1 socio-demographic characteristics:

In Table 1: How did you measure income? What is high, medium or low mean?? How did the author categorize it?

4.2 Very long and less informative table. Example: Table 2: Bivariate analysis of variables for anemia among diabetes mellitus patients in ACSH, 2018/19, (n=262);

- What type of regression model is it? Linear or logistic? The text in the table lack inconsistency font size...

-The author is has no objective or missed the aim. The title of the table says anemia but what is presented seems mean Hb.

- I don't see the advantage of this table. If needed the crude and adjusted result can be presented using one table

4.3 Table 3: Multivariable analysis of variables for Anemia in Diabetes patients

- Summary of the regression model should be presented

- I have serious reservation on the way model is fitted and what is presented here. The way OR is calculated is incorrect, please check.

- What is the importance of P-value here?

- very wide CI?? What does this show?

- The way OR is calculated is incorrect, please check. Even the 95% CI is much wider than what you presented here

- The way OR is calculated is incorrect, please check. Look at your discussion please

5. Discussion

There is major issues on the way the findings are discussed:

- The author compared two different non-comparable setting. Ex prevalence anemia in Ethiopia versus England. Look at paragraph 3

- Statistically non-significant variables were discussed as if they were. Example look at the following paragraph:"type-2 diabetic patients

have 1.7 times (AOR,

1.67, 95%CI, 0.81-3.3, P=0.17) high chance of anemia than type-1 diabetic patients"

- The justifications given are not consistent with the finding. Example "In DM, concurrent infections like

HIV, hookworm and TB are common causes of anemia "

- Or some of the justifications that were literature based were not cited. Look at paragraph 3. Ex "In this study prevalence of anemia was

found to be higher in females (11.5% vs. 6.4%), and this is due to the biological factors like menstrual loses in those who

were in their reproductive age. In addition the higher levels of androgen in males may contribute to the lower prevalence

of anemia among males."

- In general the discussion part must be revised using sound English language and should be well organized, using good scientific justification

- It should be in line with objective and result.- The OR presented is incorrect in most of the cases

- the author is expected to present the strength and limitation of the study

6 Conclusion

- Must made revised in line with update of document

7. Reference:

Please follow the guideline and correct the referencing

6. PLOS authors have the option to publish the peer review history of their article (what does this mean?). If published, this will include your full peer review and any attached files.

Reviewer #1: Yes: Dagnachew Eyachew Amare

Reviewer #2: No

---

## [Author Response · Author response to Decision Letter 0]

12 Aug 2020

Response to Reviewers

Reviewers comment response

1. The authors declare that they don’t have specific funding source. However, they claim in the acknowledgement section that they have funding source from Addis Ababa University and Adigrat. This conflict of depiction need to be resolved.

 We have changed that the funding was from Addis Ababa University (internal organization) and we corrected the cover letter also according this. But the funder had no role in study design, data collection, analysis and decision to publish or preparation of the manuscript.

2. Abstract: The authors ascertained that “anemia is higher in diabetes patients attending Ayder

Comprehensive Specialized Hospital” before the study conducted which is a wrong premise. How the author explains their claim about the existence of a high incidence of anemia among DM patients before the study conducted Vs the need to conduct the study to determine the magnitude?

In general, in the abstract part the problem statement and justification should be provided unambiguously. 

 I accept the comments and corrected accordingly.

1. In the discussion part, the authors should support “free radicals formation due to DM” with reference(s).

 Here, reference number 32 is used to serve for that whole paragraph and to make it clear and specific I add that reference for that sentence specifically.

---

## [Editor Report · Decision Letter 1]

9 Sep 2020

PONE-D-20-02501R1

The magnitude of Anemia and Associated Factors among Adult Diabetes Patients in Tertiary Teaching Hospital, Northern Ethiopia, 2019, cross-sectional study

PLOS ONE

Dear Dr. Hailu,

Thank you for submitting your manuscript to PLOS ONE. After careful consideration, we feel that it has merit but does not fully meet PLOS ONE’s publication criteria as it currently stands. Therefore, we invite you to submit a revised version of the manuscript that addresses the points raised during the review process.

Please address the comments below.

The text requires to be extensively revised by an English mother tongue expert.

TITLE

In the title, correct to …adult diabetic patients….

ABSTRACT

Background

Correct to…Patients with diabetes mellitus…

The sentence …and the current study revealed that the magnitude of anemia among diabetes patients is high…. is not appropriate here as it reports the final results.

Conclusion

Correct to “HIV status”

TEXT

Methods

State which are the Criteria for diabetes diagnosis (WHO?, other?, which year?).

We look forward to receiving your revised manuscript.

Kind regards,

Paolo Magni

Academic Editor

PLOS ONE

Additional Editor Comments (if provided):

The text requires to be extensively revised by an English mother tongue expert.

TITLE

In the title, correct to …adult diabetic patients….

ABSTRACT

Background

Correct to…Patients with diabetes mellitus…

The sentence …and the current study revealed that the magnitude of anemia among diabetes patients is high…. is not appropriate here as it reports the final results.

Conclusion

Correct to “HIV status”

TEXT

Methods

State which are the Criteria for diabetes diagnosis (WHO?, other?, which year?).

---

## [Author Response · Author response to Decision Letter 1]

28 Sep 2020

generally, the comments were found to be constructive and the really gave us a good input for our work.

thank you.

---

## [Editor Report · Decision Letter 2]

1 Oct 2020

The magnitude of Anemia and Associated Factors among Adult Diabetic Patients in Tertiary Teaching Hospital, Northern Ethiopia, 2019, cross-sectional study.

PONE-D-20-02501R2

Dear Dr. Hailu,

We’re pleased to inform you that your manuscript has been judged scientifically suitable for publication and will be formally accepted for publication once it meets all outstanding technical requirements.

Kind regards,

Paolo Magni

Academic Editor

PLOS ONE

Additional Editor Comments (optional):

All questions have been addressed.

Some further polishing of the English language is recommended.

---

## [Editor Report · Acceptance letter]

19 Oct 2020

PONE-D-20-02501R2 

The magnitude of Anemia and Associated Factors among Adult Diabetic Patients in Tertiary Teaching Hospital, Northern Ethiopia, 2019, cross-sectional study 

Dear Dr. Hailu:

I'm pleased to inform you that your manuscript has been deemed suitable for publication in PLOS ONE. Congratulations! Your manuscript is now with our production department. 

Kind regards, 

on behalf of

Prof. Paolo Magni 

Academic Editor

PLOS ONE